# INTERPRETING VIDEO FEATURES: A COMPARISON OF 3D CONVOLUTIONAL NETWORKS AND CONVOLUTIONAL LSTM NETWORKS

## ABSTRACT

A number of techniques for interpretability have been presented for deep learning in computer vision, typically with the goal of understanding what the networks have actually learned underneath a given classification decision. However, when it comes to deep video architectures, interpretability is still in its infancy and we do not yet have a clear concept of how to decode spatiotemporal features. In this paper, we present a study comparing how 3D convolutional networks and convolutional LSTM networks, respectively, learn features across temporally dependent frames. This is the first comparison of two video models that both convolve to learn spatial features but that have principally different methods of modeling time. Additionally, we extend the concept of meaningful perturbation introduced by Fong & Vedaldi (2017) to the temporal dimension to search for the most meaningful part of a sequence for a classification decision.

## 1 INTRODUCTION

Two standard approaches to deep learning for sequential image data are 3D Convolutional Neural Networks (3D CNNs), e.g. the I3D model (Carreira & Zisserman (2017)), and recurrent neural networks (RNNs). Among the RNNs, the convolutional long short-term memory network (C-LSTM) (Shi et al. (2015)) is especially suited for sequences of images, since it learns both spatial and temporal dependencies simultaneously. Although both methods can capture aspects of the semantics pertaining to the temporal dependencies in a video clip, there is a fundamental difference in how 3D CNNs treat time compared to C-LSTMs. In 3D CNNs the time axis is treated just like a third spatial axis, whereas C-LSTMs only allow for information flow in the direction of increasing time, complying with the second law of thermodynamics. More concretely, C-LSTMs maintain a hidden state representing the current video frame when forward-traversing the input video sequence, and are able to model non-linear transitions in time. 3D CNNs instead convolve (i.e. take a weighted average) over both the temporal and spatial dimensions of the sequence.

The question investigated in this paper is whether this difference has consequences for how the two models compute spatiotemporal features. We present a qualitative study of how 3D CNNs and C-LSTMs respectively compute video features: what do they learn, and how do they differ from one another?

As outlined in Section 2, there is a large body of work on evaluating video architectures on spatial and temporal correlations, but significantly fewer investigations of what parts of the data the networks have used and what semantics relating to the temporal dependencies they have extracted from it. Deep neural networks are known to be large computational models, whose inner workings are difficult to overview for a human. For video models, the number of parameters is typically significantly higher which makes their interpretability all the more pressing.

We will evaluate these two types of models (3D CNN and C-LSTM) on tasks where temporal order is crucial. The 20BN-Something-something-V2 dataset (Mahdisoltani et al. (2018), hereon Something-something) will be central to our investigations; it contains time-critical classes, agnostic to object appearance, such as *move something from left to right* or *move something from right to left*. We additionally evaluate the models on the smaller KTH actions dataset (Schuldt et al. (2004)).

Our contributions are listed as follows.

- We present the first comparison of 3D CNNs and C-LSTMs in terms of temporal modeling abilities and highlight the essential difference between their assumptions concerning temporal dependencies in the data.

- We extend the concept of meaningful perturbation introduced by Fong & Vedaldi (2017) to the temporal dimension to search for the most critical part of a sequence for classification.

## 2 RELATED WORK

The field of interpretability in the context of deep neural networks is still young but has made considerable progress for single image networks, owing to works such as Zeiler & Fergus (2013), Simonyan et al. (2014) and Montavon et al. (2018). One can distinguish between data centric and network centric methods for interpretability. *Activity maximization*, first coined by Erhan et al. (2009), is network centric in the sense that specific units of the network are being studied. By casting the maximization of the activation of a certain unit as an optimization problem in terms of the input, one can compute the optimal input for that particular unit by gradient ascent. In data centric interpretability methods, the focus is instead on the input to the network, to reveal which patterns of the data that the network has discerned.

Grad-CAM (Selvaraju et al. (2017)) and the meaningful perturbations explored in Fong & Vedaldi (2017), which form the basis for our experiments, belong to the data centric category. These two methods are further explained in Section 3. Layer-wise relevance propagation (LRP) (Montavon et al. (2018)) as well as Excitation backprop (Zhang et al. (2016)) are two other examples of data centric backpropagation techniques designed for interpretability, where the excitation backprop method follows from a simpler parameter setting of LRP. Building on excitation backprop by Zhang et al. (2016), Adel Bargal et al. (2018) produce saliency maps for video RNNs without the use of gradients. Instead, products of forward weights and activations are normalized in order to be used as conditional probabilities, which are back-propagated. We have chosen to use Grad-CAM for our experiments, since it is one of the saliency methods in Adebayo et al. (2018) that passes the article's sanity checks, as well as for its simplicity of implementation and wide usage.

Limited works have been published with their focus on interpretability for video models (Feichtenhofer et al. (2018), Sigurdsson et al. (2017), Huang et al. (2018), Ghodrati et al. (2018)). Other works have treated it, but with less extensive experimentation (Chattopadhyay et al. (2017)), while for example mainly presenting a new spatiotemporal architecture (Dwibedi et al. (2018), Zhou et al. (2018)). We build on the work by Ghodrati et al. (2018), where the aim is to measure a network's ability to model *video time* directly, instead of via the proxy task of action classification, which is most commonly seen. Three defining properties of video time are defined in the paper: temporal symmetry, temporal continuity and temporal causality, and are each presented accompanied by a measurable task. The third property is measured using the classification accuracy on the Something-something dataset. An important contribution of ours with respect to this work is that we compare 3D CNNs and C-LSTMs, whereas Ghodrati et al. (2018) compare 3D CNNs to standard LSTMs. Their comparison can be argued as slightly unfair, as standard LSTM layers only take 1D input, and thus needs to collapse each image frame in the video to a vector, which removes some spatial dependencies in the pixel grid.

Similar to our work, Dwibedi et al. (2018) investigate the temporal modeling capabilities of convolutional RNNs (Convolutional Gated Recurrent Units) trained on Something-something. The authors find that recurrent models perform well for the task, and a qualitative analysis of the learned hidden states of their trained model is presented. For each class of the dataset, they obtain the hidden states of the network corresponding to the frames of one clip and display its nearest neighbors from other clips' per-frame hidden state representations. These hidden states had encoded information about the relevant frame ordering for the classes. Sigurdsson et al. (2017) examined video architectures and datasets on a number of qualitative attributes. Huang et al. (2018) investigate how much the actual motion in a clip contributes the classification performance of a video architecture. To measure this, they perform classification experiments varying the number of sub-sampled frames used for a clip to examine how much the accuracy changes as a result.

In a search-based precursor to our temporal mask experiments, Satkin & Hebert (2010) crop sequences temporally to obtain the most discriminative sub-sequence for a certain class. The cropping corresponding to the highest classification confidence is selected as being the most discriminative sub-sequence.

Feichtenhofer et al. (2018) present the first network centric interpretability work for video models. The authors investigate spatiotemporal features using activity maximization. Zhou et al. (2018) introduce the Temporal Relational Network (TRN) which learns temporal dependencies between frames through sampling the semantically relevant frames for a particular action class. The TRN module is put on top of a convolutional layer and consists of a fully connected network between the sampled frame features and the output. Similar to Dwibedi et al. (2018), they perform temporal alignment of clips from the same class, using the frames considered most representative for the clip by the network. They verify the conclusion previously made by Xie et al. (2017), that temporal order is crucial on Something-something and show that their architecture is sensitive to that. They also investigate which classes of Something-something show the strongest sensitivity to temporal order.

# 3 APPROACH

## 3.1 TEMPORAL MASKS

The proposed temporal mask method aims to extend the interpretability of deep networks into the temporal dimension, utilizing meaningful perturbation of the input, as shown effective in the spatial dimension by Fong & Vedaldi (2017). When adopting this approach, it is necessary to define what constitutes a *meaningful* perturbation. In the mentioned paper, a mask that blurs the input as little as possible is learned for a single image, while still maximizing the decrease in class score. Our proposed method applies this concept of a learned mask to the temporal dimension. The perturbation, in this setting, is a noise mask approximating either a 'freeze' operation, which removes motion data through time, or a 'reverse' operation that inverses the sequential order of the frames. This way, we aim to identify which frames are potentially most critical for the network's classification decision.

The perturbing temporal mask is defined as a vector of values on the interval [0,1] with the same length as the input sequence. For the 'freeze' type mask, a value of 1 for a frame at index $t$ duplicates the value from the previous frame at $t-1$ onto the input sequence at $t$. The pseudocode for this procedure is given below.

**for** i in maskIndices **do**
$\quad perturbedInput_i \leftarrow (1 - mask_i) * originalInput_i + mask_i * perturbedInput_{i-1}$
**end for**

For the 'reverse' mask type, all indices of the mask **m** that are activated are first identified (threshold 0.1). These indices are then looped through to find all contiguous sections, which are treated as sub-masks, $m_i$. For each sub-mask, the frames at the active indices in the sub-mask are reversed. For example (binary for clarity), an input indexed as $t_{1:16}$ perturbed with a mask with the value $[0,0,0,1,1,1,1,1,0,0,0,0,0,1,1,0]$ would result in the sequence with frame indices $[1,2,3,8,7,6,5,4,9,10,11,12,13,15,14,16]$.

In order to learn the mask, we define a loss function (Eq. 1) to be minimized using gradient descent, similar to the approach in Fong & Vedaldi (2017).

$$\mathcal{L} = \lambda_1 \|\mathbf{m}\|_1^1 + \lambda_2 \|\mathbf{m}\|_\beta^\beta + F_c, \tag{1}$$

where **m** is the mask expressed as a vector $m \in [0,1]^t$, $\|\cdot\|_1^1$ is the $L^1$ norm, $\|\cdot\|_\beta^\beta$ is the Total Variation (TV) norm, $\lambda_{1,2}$ are weighting factors, and $F_c$ is the class score given by the model for the perturbed input. The $L^1$ norm punishes long masks, in order to identify only the most important frames in the sequence. The TV norm penalizes masks that are not contiguous.

This approach allows our method to automatically learn masks that identify one or several contiguous sequences in the input. The mask is initialized centered at the middle of the sequence. To keep the perturbed input class score differentiable w.r.t. the mask, the optimizer operates on a mask vector that has values in $\mathbb{R}$. A sigmoid function is applied to the mask before using it for the perturbing operation in order to keep its values in the [0,1] range.

The ADAM optimizer is then used to learn the mask through 300 iterations of gradient descent. After the mask has converged, it is thresholded for visualisation purposes.

## 3.2 GRAD-CAM

Grad-CAM (Selvaraju et al. (2017)) is a method for producing visual explanations in the form of class-specific saliency maps for CNNs. One saliency map, $L_t^c$, is produced for each image input based on the activations from k filters, $A_{ij}^k$, at the final convolutional layer. In order to adapt the method to sequences of images, the activations for all timesteps $t$ in the sequences are considered.

$$L_{ijt}^c = \sum_k w_k^c A_{ijt}^k \quad ; \qquad w_k^c = \frac{1}{Z} \sum_{ij} \frac{\partial F^c}{\partial A_{ijt}^k}, \tag{2}$$

where Z is a normalizing constant and $F^c$ is the network output for the class $c$. Since the aim of the method is to identify which activations had the highest contribution to the class score, only positive values of the linear combination of activations are considered, as areas with negative values are likely to belong to other classes. By up-sampling these saliency maps to the resolution of the original input image, the aim is to examine what spatial data in specific frames contributed most to the predicted class.

## 4 EXPERIMENTS

### 4.1 DATASETS

The Something-something dataset contains over 220,000 sequences from 174 classes in a resolution of 224x224 pixels. The duration of the data is more than 200 hours, and the videos were recorded against varying backgrounds from different perspectives. The classes are action-oriented and object-agnostic. Each class is defined as performing some action with one or several arbitrary objects, such as *closing something* or *folding something*. This encourages the classifier to learn the template actions, since object recognition does not give enough information for the classifying task. We train and validate according to the provided split. Fig. 2 shows sequences from the validation set.

The KTH Actions dataset consists of 25 subjects performing six different actions (*boxing, waving, clapping, walking, jogging, running*) in four different settings, resulting in a total of 2391 sequences, with a total duration of almost three hours. The videos are provided with a resolution of 160x120 pixels at 25 fps. They are filmed against a homogeneous background with the different settings exhibiting varying lighting, distance to the subject and clothing of the participants. For this dataset, we trained on subjects 1-16 and evaluated on subjects 17-25 (Fig. 3). Both datasets have sequences varying from one to almost ten seconds. As 3D CNNs require a fixed sequence length, all input sequences from both datasets were sub-sampled to cover the entire sequence in 16 frames for Something-something and 32 frames for KTH Actions. The same set of sub-sampled frames was then used as input to each architecture for each dataset.

### 4.2 ARCHITECTURES AND EXPERIMENT DETAILS

Hyperparameters are listed in the appendix. Any remaining settings can be found in the code which will be made public in both Pytorch and Tensorflow.

I3D consists of three 3D convolutional layers, nine Inception modules and four max pooling layers (Fig. 1). In the original setting, the temporal dimension of the output is down-sampled to two frames. In order to achieve a higher temporal resolution in the produced Grad-CAM images, the strides of the first convolutional layer and the second max pooling layer were reduced to 1x2x2, producing eight activations in the temporal dimension for the 16 frame inputs. The Grad-CAM images are produced from the gradients of the class scores w.r.t. the final Inception module.

The C-LSTM architecture used for Something-something consisted of three C-LSTM layers, each followed by batch normalization and max pooling layers. The convolutional kernels used for each layer had size 5x5 and stride 2x2 with 32 filters. The C-LSTM layers return the entire transformed sequence as input to the next layer, including the last layer before the fully connected layer, used for

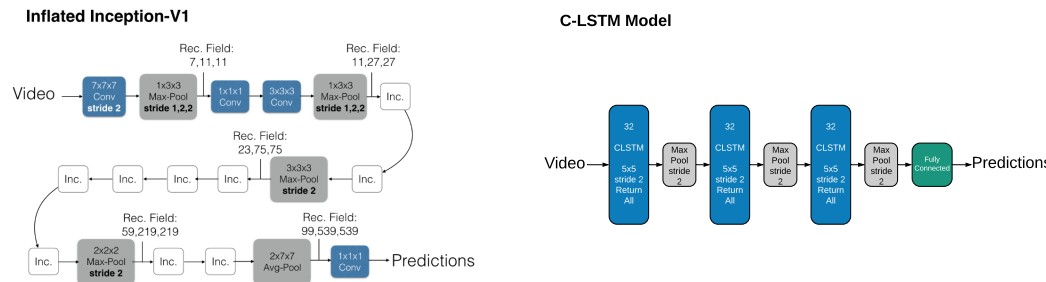

Figure 1: I3D network (figure from Carreira & Zisserman (2017)) and C-LSTM network (right).

the predictions. For KTH, the C-LSTM model had two layers with 32 hidden units each and dropout between the layers ($p = 0.5$). These architectures were chosen as the best performing models after empirical experimentation with the number of layers, hidden units, stride and regularization. When calculating the Grad-CAM maps for the C-LSTM, the final C-LSTM layer was used.

We note that there is a substantial difference in the number of parameters for each resulting model, with $12,465,614$ for I3D and $1,324,014$ and for the three-layer C-LSTM. When introduced, the I3D architecture achieved state-of-the-art performance on several video recognition datasets. These properties combined suggest that I3D should have an advantage in performance over the two models. This was confirmed on Something-something, where the C-LSTM architecture could not reach the same overall performance as I3D (Table 1). Other architectural variants of the C-LSTM model with a larger amount of parameters (up to five layers) were evaluated as well, but no significant increase in performance was observed. Also, due to the computational complexity of back-propagation through time (BPTT), the C-LSTM variants were significantly more time demanding to train and evaluate than their I3D counterparts. With this in mind, in order to make the comparison as fair as possible, classes were chosen for which the performance of the two architectures were similar. The labels of these classes and their F1 scores for each architecture are shown in Section 5.1 and in the appendix.

## 5 RESULTS

The F1-scores for both architectures and datasets are shown in Table 1. In order to investigate how reliant the two models are on the temporal order of the input frames, a further test was conducted with the input sequences reversed. On Something-something, both the C-LSTM and I3D models were affected drastically, with their top-1 classification scores dropping by 78% and 79%, respectively. This suggests that both models are in fact sensitive to the temporal direction, to almost the same degree, when the test sequence is entirely reversed. In Sections 5.1-5.2, we present results for when only the most salient portion of a sequence is reversed.

For both models, the highest scoring class after reversing the sequence was *turning something upside down*. This is perhaps not surprising, as the semantic meaning of the action holds even when played backward. The classes with the largest drop in score after reversal for both models were those containing movement in a specific direction, such as *turning the camera left while filming something* or *pushing something from left to right*. Both models performed well when reversing the KTH Actions dataset. Most likely, this is because this dataset's classes have distinct spatial features.

Table 1: F1-score of each model on the datasets KTH Actions and Something-something.

| Model | KTH Actions (Top-1) | Smth-Smth (Top-1) | Smth-Smth (Top-5) |
|---|---|---|---|
| C-LSTM | 0.84 | 0.23 | 0.48 |
| C-LSTM (reversed) | 0.78 | 0.05 | 0.17 |
| I3D | 0.86 | 0.43 | 0.73 |
| I3D (reversed) | 0.80 | 0.09 | 0.27 |

## 5.1 INTERPRETABILITY RESULTS ON SOMETHING-SOMETHING

In this section, we present the Grad-CAM heatmaps and temporal masks generated for each architecture. We display eight sequences in Fig. 2, but have included more examples in the appendix. The classes of this section's analysis are as follows (I3D F1/C-LSTM F1): *moving something and something away from each other* (0.76/0.58), *moving something and something closer to each other* (0.77/0.57), *moving something and something so they pass each other* (0.37/0.31), *moving something up* (0.43/0.4), *pretending to take something from somewhere* (0.1/0.07), *moving the camera down while filming something* (0.67/0.56), and *moving the camera up while filming something* (0.81/0.73).

First, we note that the Something-something classes can be ambiguous (one class may contain another class) and for a few samples, arguably, even be incorrectly labeled. The latter can be seen for example in Sequence #2, where I3D's classification was *moving something and something so they collide with each other* and the C-LSTM model predicted *pushing something with something*. Although the two objects in the sequence do move closer to each other, they also touch at the very end, making both predictions technically correct. Another case of understandable confusion can be seen in Sequence #5, where I3D's classification was *taking one of many similar things on the table*. In this case the surface seen in the image is a tiled floor, and the object is a transparent ruler. Once the temporal mask activates during the lifting motion in the last four frames, the Grad-CAM images show that the model also focuses on two of the lines on the floor. These could be considered similar to the lines caused by the outline of the ruler, which could explain the incorrect classification.

A difference between the architectures is that the I3D model often focuses on contiguous, centered blobs, while the C-LSTM model attempts to find relevant spatial features in multiple smaller areas (Table 2). Examples can be seen in Sequences #1 and #3 of Fig. 2, where I3D focuses on a single region covering both objects while the C-LSTM has activations for both of the objects and the surface affected by the movement. The I3D model also has a bias of starting its focus around the middle of the screen as can be seen in Sequences #1 to #8, often even before the motion starts. The typical behavior for C-LSTM is instead to remain agnostic until the action actually starts (Sequence #7). For Sequence #7, the I3D maintains its foveal focus even after the green, round object is out of frame. For Sequence #8, the focus actually splits midway to cover both the moped and some features on the wall, while the C-LSTM model focuses mainly on numerous features along the wall, as it usually does in classes where the camera turns. The C-LSTM also seems to pay more attention to hands appearing in the clips, rather than objects, as can be seen in Sequences #1 to #4. The average distance to the center of the image for each blob in each frame is shorter for I3D (Table 2).

Given the same number of iterations for the optimization of the temporal mask, the two models typically reached different losses. Generally, I3D obtained a lower loss. For this reason, we consider the difference and ratio[1] between the score drops after the freeze and reverse perturbations as the most relevant measures of how sensible one model was for the reverse perturbation. We observe that, in general, the drop caused by the reverse perturbation is smaller for the C-LSTM than for I3D. However, the freeze-reverse drops' difference and ratio are considerably higher for the I3D compared to C-LSTM (Table 2), suggesting that I3D is less sensitive to the salient reverse perturbation.

Last, we note that the average temporal mask is shorter for the I3D model (Table 2). This suggests that it has learned to react to short, specific events in the sequences. As an example, the temporal mask of Sequence #3 is active only on the frames where the objects first pass each other, and in Sequence #2, it is active on the frames leading to the objects touching.

OS: 0.994                                                                                        OS: 0.312
FS: 0.083                                                                                        FS: 0.186
RS: 0.856                                                                                        RS: 0.125

Sequence #1: Moving something and something away from each other.

---

[1]For the drop ratio, if the denominator OS-RS $\leq 0.01$, the sample was filtered out since its ratio would explode. The OS-FS $\leq 0.01$ were also excluded for balance. Without any filtering, the results were $323 \pm 8640$ (I3D) and $22.0 \pm 588$ (C-LSTM).

Table 2: Statistics for the temporal masks and Grad-CAM maps on a subset of the Something-something validation set (791 samples). The analysis was run on 150 clips each (fewer if there were not 150 samples of that class in the validation set) of the seven classes presented in this section. The 'blobs' were computed per frame, using the blob detection tool from OpenCV (Bradski (2000)). Drop ratio is defined as $\frac{OS-FS}{OS-RS}$ and drop difference as $(OS - FS) - (OS - RS)$ (Fig. 2).

|  | Mask length | Drop ratio | Drop diff. | Blob count | Blob size | Center distance |
|---|---|---|---|---|---|---|
| I3D | $6.1 \pm 3.3$ | $3.7 \pm 9.7$ | $0.2 \pm 0.3$ | $1.5 \pm 0.35$ | $33.9 \pm 19.7$ | $54.8 \pm 32.9$ |
| C-LSTM | $9.6 \pm 4.2$ | $2.1 \pm 3.9$ | $0.08 \pm 0.2$ | $3.6 \pm 0.6$ | $25.9 \pm 23.4$ | $96.8 \pm 34.7$ |

OS: 0.547
FS: 0.028
RS: 0.053
CS: 0.186
P: 38

OS: 0.257
FS: 0.079
RS: 0.122
CS: 0.002
P: 135

Sequence #2: Moving something and something closer to each other.

OS: 0.999
FS: 0.002
RS: 0.414

OS: 0.788
FS: 0.392
RS: 0.537

Sequence #3: Moving something and something so they pass each other.

OS: 0.804
FS: 0.016
RS: 0.667

OS: 0.546
FS: 0.121
RS: 0.764

Sequence #4: Moving something up.

OS: 0.685
FS: 0.003
RS: 0.048
CS: 0.001
P: 146

OS: 0.221
FS: 0.182
RS: 0.350
CS: 0.005
P: 100

Sequence #5: Moving something up.

OS: 0.284
FS: 0.003
RS: 0.006

OS: 0.600
FS: 0.167
RS: 0.088
CS: 0.004
P: 27

Sequence #6: Pretending to take something from somewhere.

OS: 1.000
FS: 0.001
RS: 0.011

OS: 0.158
FS: 0.063
RS: 0.093

Sequence #7: Turning the camera downwards while filming something.

OS: 0.990
FS: 0.001
RS: 0.000

OS: 0.806
FS: 0.177
RS: 0.181

Sequence #8: Turning the camera upwards while filming something.

Figure 2: **Best displayed in Adobe Reader where the figures can be played as videos.** *I3D (left) and C-LSTM (right) results for validation sequences from Something-something. The three columns show, from left to right, the original input, the Grad-CAM result, and the input as perturbed by the temporal freeze mask. The third column also visualizes when the mask is on (red) or off (green), with the current frame highlighted. OS: original score (softmax output) for the guessed class, FS: freeze score, RS: reverse score and CS: score for the ground truth class when there was a misclassification.*

## 5.2 Interpretability results on the KTH actions dataset

In Fig. 3, we observe results for the class 'handclapping'. Interestingly, the mask of each model covers at least one entire cycle of the action. The mask is smaller for C-LSTM and for that reason does not lower its score as much as for I3D, whose freeze score is very low compared to the original and reverse score. This can be further explained by watching the frozen sequence and observing that no full cycle remains from the action. The reverse perturbation affects both models very little since one action cycle is symmetrical in time. For the 'running' class, we see that the temporal mask identifies the frames in which the subject is in-frame as the most salient for both models. However, the Grad-CAM results show that the I3D model places more focus on the subject's legs than the C-LSTM version. This is also reflected in the temporal mask for I3D, which activates first when it has started to shift its focus to the legs.

OS: 0.999                                                                                               OS: 0.996
FS: 0.026                                                                                               FS: 0.997
RS: 0.999                                                                                               RS: 0.996

Handclapping, subject 18.

OS: 0.993                                                                                               OS: 0.669
FS: 0.208                                                                                               FS: 0.339
RS: 0.999                                                                                               RS: 0.605

Running, subject 25.

Figure 3: **The figures can be displayed as videos in Adobe Reader.** *Same structure as Fig. 2.*

## 6 CONCLUSIONS AND FUTURE WORK

In this work we have presented a comparison of the spatiotemporal information used by 3D CNN and C-LSTM based models to perform video classification on two datasets, aiming to narrow down what they learn, and how they differ from one another. We analyzed the spatial information used by each model using the Grad-CAM method, and proposed the temporal mask method to investigate which video segments are most important for the classification. The comparison suggests that the 3D CNN on average focuses on shorter and more specific sequences than the C-LSTM model. On average, the 3D CNN also tends to focus on fewer or a single contiguous spatial patch, instead of smaller areas on several objects like the C-LSTM. Also, when comparing the effect of removing motion either through 'freezing' the most salient frames or reversing their order, the C-LSTM experiences a relatively higher decrease in prediction confidence than I3D upon reversal. We have also seen that the proposed temporal mask is capable of identifying salient frames in sequences, such as one cycle of a repetitive motion, or the instance of a passing motion.

There is still much to explore in the patterns lying in temporal dependencies. The compared architectures had a difference in performance on the more difficult Something-something dataset. If an established C-LSTM architecture that performs equally well becomes available in the future, it would be of interest to revisit this comparison. Likewise, it would be of interest to extend the study to other datasets where temporal information is important, e.g. Charades (Sigurdsson et al. (2016)). Other possible future work includes evaluating the effect of other noise types beyond 'freeze' and 'reverse'. We also believe that in the future it would be of interest to gain further insight into state-of-the-art models performing video classification benchmarks by utilizing the proposed tools.

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

## A  APPENDIX

### A.1  TRAINING HYPERPARAMETERS

Table 3: Hyperparameters used for training the two models on each dataset.

| Model (Dataset) | Hyperparameter | | | | |
|---|---|---|---|---|---|
| | Dropout Rate | Weight Decay | Optimizer | Epochs | Momentum |
| I3D (Smth-smth) | 0.5 | 0 | ADAM | 13 | - |
| I3D (KTH) | 0.7 | 5E-5 | ADAM | 30 | - |
| C-LSTM (Smth-smth) | 0.0 | 0 | SGD | 105 | 0.2 |
| C-LSTM (KTH) | 0.5 | 1E-4 | SGD | 21 | 0.2 |

Table 4: Hyperparameters used for optimizing the temporal mask.

| Dataset | Hyperparameter | | | | |
|---|---|---|---|---|---|
| | $\lambda_1$ | $\lambda_1$ | Optimizer | Iterations | Learning rate |
| Smth-smth | 0.01 | 0.02 | ADAM | 300 | 0.001 |
| KTH | 0.02 | 0.04 | ADAM | 300 | 0.001 |

### A.2  FURTHER SEQUENCE EXAMPLES: SOMETHING-SOMETHING

Below, we present results for 22 additional randomly selected sequences (two from each class) from the Something-something dataset. As mentioned in the main article, we selected eleven classes where the two models had comparable performance. The four classes not appearing above are (I3D F1-score/C-LSTM F1 score): *moving something and something so they collide with each other* (0.16/0.03), *burying something in something* (0.1/0.06), *turning the camera left while filming something* (0.94/0.79) and *turning the camera right while filming something* (0.91/0.8).

OS: 1.000                                                                OS: 0.373
FS: 0.015                                                                 FS: 0.547
RS: 0.003                                                                RS: 0.224

Sequence #9: Turning the camera downwards while filming something.

OS: 0.999
FS: 0.000
RS: 0.000

OS: 0.921
FS: 0.238
RS: 0.460

Sequence #10: Turning the camera downwards while filming something.

OS: 0.997
FS: 0.012
RS: 0.014

OS: 0.988
FS: 0.183
RS: 0.105

Sequence #11: Turning the camera left while filming something.

OS: 0.999
FS: 0.001
RS: 0.451

OS: 0.985
FS: 0.229
RS: 0.094

Sequence #12: Turning the camera left while filming something.

OS: 0.192
FS: 0.261
RS: 0.140
CS: 0.103
P: 157

OS: 0.940
FS: 0.106
RS: 0.017

Sequence #13: Turning the camera right while filming something.

OS: 0.947
FS: 0.005
RS: 0.188

OS: 0.708
FS: 0.093
RS: 0.119

Sequence #14: Turning the camera right while filming something.

OS: 0.999
FS: 0.001
RS: 0.002

OS: 0.687
FS: 0.205
RS: 0.149

Sequence #15: Turning the camera upwards while filming something.

OS: 0.997
FS: 0.002
RS: 0.064

OS: 0.689
FS: 0.108
RS: 0.129

Sequence #16: Turning the camera upwards while filming something.

OS: 0.297
FS: 0.155
RS: 0.294
CS: 0.085
P: 121

OS: 0.917
FS: 0.058
RS: 0.071

Sequence #17: Moving something and something away from each other.

OS: 0.991
FS: 0.022
RS: 0.956

OS: 0.259
FS: 0.081
RS: 0.146
CS: 0.008
P: 130

Sequence #18: Moving something and something away from each other.

OS: 0.273
FS: 0.004
RS: 0.245
CS: 0.001
P: 173

OS: 0.123
FS: 0.375
RS: 0.200
CS: 0.012
P: 100

Sequence #19: Moving something and something closer to each other something.

OS: 0.932
FS: 0.002
RS: 0.007

OS: 0.453
FS: 0.063
RS: 0.198

Sequence #20: Moving something and something closer to each other something.

OS: 0.686
FS: 0.003
RS: 0.000

OS: 0.620
FS: 0.145
RS: 0.129

Sequence #21: Moving something and something so they collide with each other.

OS: 0.810
FS: 0.055
RS: 0.419

OS: 0.333
FS: 0.119
RS: 0.276
CS: 0.030
P: 37

Sequence #22: Moving something and something so they collide with each other.

OS: 0.997
FS: 0.007
RS: 0.974

OS: 0.737
FS: 0.490
RS: 0.140

Sequence #23: Moving something and something so they pass each other.

OS: 0.694
FS: 0.010
RS: 0.003
CS: 0.273
P: 37

OS: 0.813
FS: 0.227
RS: 0.830
CS: 0.142
P: 37

Sequence #24: Moving something and something so they pass each other.

OS: 0.619
FS: 0.010
RS: 0.216
CS: 0.020
P: 145

OS: 0.1298
FS: 0.079
RS: 0.262
CS: 0.001
P: 157

Sequence #25: Burying something in something.

OS: 0.177
FS: 0.007
RS: 0.130
CS: 0.027
P: 106

OS: 0.112
FS: 0.147
RS: 0.327
CS: 0.002
P: 5

Sequence #26: Burying something in something.

OS: 0.848
FS: 0.065
RS: 0.380
CS: 0.003
P: 27

OS: 0.229
FS: 0.102
RS: 0.269
CS: 0.003
P: 100

Sequence #27: Moving something up.

OS: 0.230
FS: 0.146
RS: 0.200
CS: 0.033
P: 100

OS: 0.755
FS: 0.012
RS: 0.032

Sequence #28: Moving something up.

OS: 0.810
FS: 0.019
RS: 0.682
CS: 0.000
P: 160

OS: 0.179
FS: 0.073
RS: 0.162
CS: 0.004
P: 160

Sequence #29: Pretending to take something from somewhere.

OS: 0.325
FS: 0.012
RS: 0.126
CS: 0.047
P: 145

OS: 0.418
FS: 0.062
RS: 0.266
CS: 0.011
P: 160

Sequence #30: Pretending to take something from somewhere.

### A.3  FURTHER KTH ACTION EXAMPLES

Below, we present results for six additional randomly selected sequences (one from each class) from the KTH Actions dataset.

OS: 1.000
FS: 1.000
RS: 1.000

OS: 0.973
FS: 0.978
RS: 0.973

Boxing, subject 17.

OS: 0.946
FS: 0.009
RS: 0.982

OS: 0.715
FS: 0.507
RS: 0.765

Hand clapping, subject 17.

OS: 1.000
FS: 0.012
RS: 1.000

OS: 0.896
FS: 0.349
RS: 0.761

Hand waving, subject 18.

OS: 0.662
FS: 0.354
RS: 0.706
CS: 0.172
P: 4

OS: 0.991
FS: 0.353
RS: 0.002

Jogging, subject 25.

OS: 1.000
FS: 0.025
RS: 1.000

OS: 0.524
FS: 0.274
RS: 0.932

Running, subject 24.

OS: 1.000
FS: 0.010
RS: 0.775

OS: 0.724
FS: 0.333
RS: 0.654

Walking, subject 25.

## B  SOMETHING-SOMETHING CLASSES

"Approaching something with your camera":"0", "Attaching something to something":"1", "Bending something so that it deforms":"2", "Bending something until it breaks":"3", "Burying something in something":"4", "Closing something":"5", "Covering something with something":"6", "Digging something out of something":"7", "Dropping something behind something":"8", "Dropping something in front of something":"9", "Dropping something into something":"10", "Dropping something next to something":"11", "Dropping something onto something":"12", "Failing to put something into something because something does not fit":"13", "Folding something":"14", "Hitting something with something":"15", "Holding something":"16", "Holding something behind something":"17", "Holding something in front of something":"18", "Holding something next to something":"19", "Holding something over something":"20", "Laying something on the table on its side, not upright":"21", "Letting something roll along a flat surface":"22", "Letting something roll down a slanted surface":"23", "Letting something roll up a slanted surface, so it rolls back down":"24", "Lifting a surface with something on it but not enough for it to slide down":"25", "Lifting a surface with something on it until it starts sliding down":"26", "Lifting something up completely without letting it drop down":"27", "Lifting something up completely, then letting it drop down":"28", "Lifting something with something on it":"29", "Lifting up one end of something without letting it drop down":"30", "Lifting up one end of something, then letting it drop down":"31", "Moving away from something with your camera":"32", "Moving part of something":"33", "Moving something across a surface until it falls down":"34", "Moving something across a surface without it falling down":"35", "Moving something and something away from each other":"36", "Moving something and something closer to each other":"37", "Moving something and something so they collide with each other":"38", "Moving something and something so they pass each other":"39", "Moving something away from something":"40", "Moving something away from the camera":"41", "Moving something closer to something":"42", "Moving something down":"43", "Moving something towards the camera":"44", "Moving something up":"45", "Opening something":"46", "Picking something up":"47", "Piling something up":"48", "Plugging something into something":"49", "Plugging something into something but pulling it right out as you remove your hand":"50", "Poking a hole into some substance":"51", "Poking a hole into something soft":"52", "Poking a stack of something so the stack collapses":"53", "Poking a stack of something without the stack collapsing":"54", "Poking something so it slightly moves":"55", "Poking something so lightly that it doesn't or almost doesn't move":"56", "Poking something so that it falls over":"57", "Poking something so that it spins around":"58", "Pouring something into something":"59", "Pouring something into something until

it overflows":"60", "Pouring something onto something":"61", "Pouring something out of something":"62", "Pretending or failing to wipe something off of something":"63", "Pretending or trying and failing to twist something":"64", "Pretending to be tearing something that is not tearable":"65", "Pretending to close something without actually closing it":"66", "Pretending to open something without actually opening it":"67", "Pretending to pick something up":"68", "Pretending to poke something":"69", "Pretending to pour something out of something, but something is empty":"70", "Pretending to put something behind something":"71", "Pretending to put something into something":"72", "Pretending to put something next to something":"73", "Pretending to put something on a surface":"74", "Pretending to put something onto something":"75", "Pretending to put something underneath something":"76", "Pretending to scoop something up with something":"77", "Pretending to spread air onto something":"78", "Pretending to sprinkle air onto something":"79", "Pretending to squeeze something":"80", "Pretending to take something from somewhere":"81", "Pretending to take something out of something":"82", "Pretending to throw something":"83", "Pretending to turn something upside down":"84", "Pulling something from behind of something":"85", "Pulling something from left to right":"86", "Pulling something from right to left":"87", "Pulling something onto something":"88", "Pulling something out of something":"89", "Pulling two ends of something but nothing happens":"90", "Pulling two ends of something so that it gets stretched":"91", "Pulling two ends of something so that it separates into two pieces":"92", "Pushing something from left to right":"93", "Pushing something from right to left":"94", "Pushing something off of something":"95", "Pushing something onto something":"96", "Pushing something so it spins":"97", "Pushing something so that it almost falls off but doesn't":"98", "Pushing something so that it falls off the table":"99", "Pushing something so that it slightly moves":"100", "Pushing something with something":"101", "Putting number of something onto something":"102", "Putting something and something on the table":"103", "Putting something behind something":"104", "Putting something in front of something":"105", "Putting something into something":"106", "Putting something next to something":"107", "Putting something on a flat surface without letting it roll":"108", "Putting something on a surface":"109", "Putting something on the edge of something so it is not supported and falls down":"110", "Putting something onto a slanted surface but it doesn't glide down":"111", "Putting something onto something":"112", "Putting something onto something else that cannot support it so it falls down":"113", "Putting something similar to other things that are already on the table":"114", "Putting something that can't roll onto a slanted surface, so it slides down":"115", "Putting something that can't roll onto a slanted surface, so it stays where it is":"116", "Putting something that cannot actually stand upright upright on the table, so it falls on its side":"117", "Putting something underneath something":"118", "Putting something upright on the table":"119", "Putting something, something and something on the table":"120", "Removing something, revealing something behind":"121", "Rolling something on a flat surface":"122", "Scooping something up with something":"123", "Showing a photo of something to the camera":"124", "Showing something behind something":"125", "Showing something next to something":"126", "Showing something on top of something":"127", "Showing something to the camera":"128", "Showing that something is empty":"129", "Showing that something is inside something":"130", "Something being deflected from something":"131", "Something colliding with something and both are being deflected":"132", "Something colliding with something and both come to a halt":"133", "Something falling like a feather or paper":"134", "Something falling like a rock":"135", "Spilling something behind something":"136", "Spilling something next to something":"137", "Spilling something onto something":"138", "Spinning something so it continues spinning":"139", "Spinning something that quickly stops spinning":"140", "Spreading something onto something":"141", "Sprinkling something onto something":"142", "Squeezing something":"143", "Stacking number of something":"144", "Stuffing something into something":"145", "Taking one of many similar things on the table":"146", "Taking something from somewhere":"147", "Taking something out of something":"148", "Tearing something into two pieces":"149", "Tearing something just a little bit":"150", "Throwing something":"151", "Throwing something against something":"152", "Throwing something in the air and catching it":"153", "Throwing something in the air and letting it fall":"154", "Throwing something onto a surface":"155", "Tilting something with something on it slightly so it doesn't fall down":"156", "Tilting something with something on it until it falls off":"157", "Tipping something over":"158", "Tipping something with something in it over, so something in it falls out":"159", "Touching (without moving) part of something":"160", "Trying but failing to attach something to something because it doesn't stick":"161", "Trying to bend something unbendable so nothing happens":"162", "Trying to pour something into something, but missing so it spills next to it":"163", "Turning something upside down":"164", "Turning the camera downwards

while filming something":"165", "Turning the camera left while filming something":"166", "Turning the camera right while filming something":"167", "Turning the camera upwards while filming something":"168", "Twisting (wringing) something wet until water comes out":"169", "Twisting something":"170", "Uncovering something":"171", "Unfolding something":"172", "Wiping something off of something":"173"

## C  KTH ACTIONS CLASSES

"Boxing": "0", "Handclapping": "1", "Handwaving":"2", "Jogging":"3", "Running":"4", "Walking":"5"

