# OpenReview forum: "Interpreting video features: a comparison of 3D convolutional networks and convolutional LSTM networks"
_ICLR.cc/2020/Conference — Reject_

### Official Review · AnonReviewer1 · 2019-10-11
**Official Blind Review #1**

**Rating:** 6

**Review:**

The paper shows a way to compare what is learned by two very different networks trained for a video classification task. The two architectures are state-of-the-art methods, one relying on 3d-CNNs (time= one dimension), the other on conv-LSTMs (time is treated sequentially, using hidden states to pass information). The idea of the authors is (i) to provide saliency maps for each of them, and (ii) to create interesting perturbations in order to measure the influence on the networks. The results indicate that these complex networks are usually focused on interesting features, and as we would imagine, LSTMs is more learning from temporal coherence than CNNs.

Overall, I think the paper is worth to be published at ICLR, even if I am not aware of the recent publications in this field. The contribution in terms of method is small, but I think that such careful studies can be very fruitful to the community.

Positive aspects:
- A significant effort has been put to creating meaningful perturbations for this particular task, i.e. temporal dependance and coherence.
- The effort to compare as best as possible two different approaches is fruitful, and very useful for the community as this is a real question to be raised.
- The experiments are made also with care, on 2 different datasets, and large efforts were made on explaining the different results.

Questions/remarks:
- As I was under Linux, I did not manage to view the videos, and I have not seen a link for downloading them. It would be best to add it as supplementary materials?
- What do you mean by 'The TV norm penalizes masks that are not coherent'? what is coherent?
- '...the mask is defined as a vector of values between [0,1]' : I understood that it was a real value between 0 and 1, but I think you meant more 'a binary vector'?
- Have you looked at the importance of the sub-sampling in the CNN framework? i.e., since the LSTM framework does not have that, maybe the differences in activation also depend on the length of the sequence. Maybe the large sequences, where CNN and LSTM would have the same number of frames, are less different?
- You are saying that the code will be made public, is it possible to make it public now?

Small remarks:
- '3D CNNs instead instead convolve'
- 'As outlined in section 2' when we are in the introduction
- '(Mahdisoltani et al.(2018) contains...'. parenthesis.
- Figure 1: the first figure is in too poor quality.


**Experience Assessment:**

I have read many papers in this area.

**Review Assessment: Checking Correctness Of Derivations And Theory:**

I assessed the sensibility of the derivations and theory.

**Review Assessment: Checking Correctness Of Experiments:**

I carefully checked the experiments.

**Review Assessment: Thoroughness In Paper Reading:**

I read the paper at least twice and used my best judgement in assessing the paper.

---

> ### Author Response · Authors · 2019-11-12
> **Reply to Official Blind Review #1**
>
> Dear reviewer,
>
> Thank you for your encouraging and insightful assessment! In the following, we address the rest of your questions and remarks one by one.
>
> - After the double-blind period, our plan is to make a website for the article displaying all the video material in the browser. Until then, unfortunately one needs to use Adobe reader to be able to watch the videos. We realize that this is not the most practical solution; however, Adobe should be available for Linux.
>
> - The word ‘coherent’ has been replaced by the more precise word ‘contiguous’ in the updated version of the article.
>
> - The code was anonymously made public on OpenReview at the same time that we submitted, see the Dropbox link along with our submission. After the double-blind period, we will release a public repository in our names.
>
> - Regarding the vector of values in [0,1], we do in effect mean that the vector can take any real values in the [0,1] interval. The reason for is that the mask needs to be differentiable. This is explained at the end of Section 3.1, but we have clarified this point throughout Sect. 3.1 in the updated version of the article and we would like to thank you for pointing to this confusion.
>
> - We also want to thank the reviewer for the valuable comment regarding the sub-sampling of CNN models. I3D does indeed subsample the sequence temporally along its forward pass, going from 16 time steps to 8 time steps in the output. It would be interesting to include results from a model with no temporal subsampling and study the characters of its temporal masks. This was unfortunately out of scope for this rebuttal period but will keep this in mind for future work.
>
> Last, we have corrected the remaining four remarks at the end of your review in the updated version of the article, and we thank the reviewer for spotting these mistakes.

---

### Official Review · AnonReviewer2 · 2019-10-22
**Official Blind Review #2**

**Rating:** 1

**Review:**

This work attempts to experimentally investigate the difference between Conv3Ds and ConvLSTMs with the aid of visualization. The visualization focuses on two aspects: (1) Temporal masking for identifying key frames and key segments and (2) GradCAM for spatial saliency. Both visualization techniques are illustrated on two public available video classification datasets.

This work has some major issues:
1. The quantitative results are not very relevant. It does not validate any of the following cases: (1) if Conv3Ds are more powerful, one should use the same number of parameters and compare classification results, or (2), in order to achieve the same level of accuracy, one of the two models is more parameter efficient. It is not clear which argument Table 1 is validating against if there is any as both parameters and accuracies vary.
2. Qualitative results from Section 5.2 is not substantiated and underwhelming. For instance, it is hard to see why "Conv3Ds has a bias around the center while ConvLSTMs find relevant spatial features in multiple smaller areas". This is most likely due to visualization techniques other than the choice of models.
3. The fundamental issue of this work is that it does not establish a hypothesis from the beginning and design experiments around it. Plenty of hand-wavy observations are made without further investigating the root of them, leaving readers unsatisfied, and quite often confused.

**Experience Assessment:**

I have published one or two papers in this area.

**Review Assessment: Checking Correctness Of Derivations And Theory:**

I assessed the sensibility of the derivations and theory.

**Review Assessment: Checking Correctness Of Experiments:**

I assessed the sensibility of the experiments.

**Review Assessment: Thoroughness In Paper Reading:**

I read the paper at least twice and used my best judgement in assessing the paper.

---

> ### Author Response · Authors · 2019-11-12
> **Reply to Official Blind Review #2**
>
> Dear reviewer,
>
> Thank you for your valuable feedback. Below, we will address each of your raised issues:
>
> 1. Regarding the fairness of the comparison, it is written at the end of section 4.2:
> 	“(...) Also, due to the computational complexity of backpropagation through time (BPTT), the C-LSTM variants were significantly more time demanding to train and evaluate than their I3D counterparts. With this in mind, in order to make the comparison as fair as possible, eleven classes were chosen for which the performance of the two architectures were similar.”
> And in Section 5.2:
> 	“The chosen classes were as follows (I3D F1/C-LSTM F1): moving something and something away from each other (0.76/0.58), moving something and something closer to each other (0.77/0.57), moving something and something so they pass each other (0.37/0.31), moving something up (0.43/0.4), pretending to take something from somewhere (0.1/0.07), moving the camera down while filming something (0.67/0.56), and moving the camera up while filming something (0.81/0.73).”
> We agree with you that this situation is not ideal since I3D has a slight overhand on the C-LSTM for each class. Even so, we decided to pursue this track for three reasons:
>
> i. We think that the difference in accuracy per class for the two models is within an acceptable range and judge that they should be at least comparable.
>
> ii. 3D CNNs clearly represent the current state-of-the-art for action recognition. Because of this, it is interesting to study what such a model — better performing in terms of accuracy —  finds, compared to a weaker one (in terms of accuracy), since this reflects a typical real-world situation (where the 3D CNN typically would be the model with highest accuracy). Furthermore, there is an argument to be made concerning how ‘good’ it is to score a high accuracy on a quite artificial dataset such as Something-something. See also a slightly longer discussion of this point at the end of our response to Official Blind Review #5 (paragraph starting with “In the reviewer’s fourth point,...”).
>
> iii. As you correctly point out, the other alternative would be to compare two models on the basis of having the same number of parameters. However, this is simply not feasible for the two concerned types of models (C-LSTM and 3D CNN). I3D has 12 times as many parameters as the C-LSTM, yet the C-LSTM takes roughly 1.5 times longer to train for one epoch (2.2h vs. 1.5h on Something-something). To train a C-LSTM with the same number of parameters as the I3D would simply take a nonviable amount of time. It can be noted that the C-LSTM used for these experiments on Something-something was trained for 229 hours (105 epochs) on an Nvidia RTX 2080 TI .
>
>
> 2. We are not quite sure what is meant by ‘This is likely due to visualization techniques other than the choice of models’ since the exact same visualization technique is applied to the two models. In what concerns the first comment about our qualitative results being unsubstantiated, we have included more extensive quantitative results in Table 2 of the updated article,  including a global measure of the number of ‘blobs’ (contiguous regions present per frame in the Grad-CAM saliency maps), their average size and distance from the center of the image, average mask length and average difference and ratio for the drop in classification score after freeze and reverse perturbations, respectively. This analysis was done across a subset of the validation data for each model (see the first two paragraphs in our response to Official Blind Review #5). The results are in line with our findings concerning the center preference of the focus of I3D compared to the C-LSTM.
>
> 3. Deliberately, we did not want to establish a hypothesis from the beginning, in order to investigate this question in an unbiased manner. When it comes to investigating the root of some claims made in the discussion section of the original version of the article, we have now substantiated these claims in the new Table 2 of the updated article. We have also improved the discussion using these results and hope that this altogether can help to diminish the reviewer’s confusion.
>
> Please don’t hesitate to reply to this if you have further questions to us at this point.

---

### Official Review · AnonReviewer5 · 2019-11-04
**Official Blind Review #5**

**Rating:** 3

**Review:**

This paper presents a paradigm for generating saliency maps for video models, specifically, I3D (3D CNN) and C-LSTM. It extends Fong & Vedaldi, 2017 to generate a temporal mask and introduces two types of "meaningful perturbations" for videos: freezing and reversing frames; they use Grad-CAM (with no modifications) for generating spatial masks. The problem is well-motivated, as saliency maps have been extensively studied for image classification models, but rarely for video classification. Quantitatively, they demonstrate that frame-reversal is meaningful for the Something-something dataset but less for KTH because those actions rely more on spatial information than temporal (i.e., running, clapping). Qualitatively, they show their spatial and temporal masks on both datasets and suggest the a few insights:
* I3D's Grad-CAM visualizations show a center, default bias * I3D is less sensitive to the reverse perturbation * I3D temporal masks are typically shorter.

I currently rate this paper as a weak reject (though closer to borderline) paper for the following reasons (all of which can be improved in rebuttal):
1. Quality of technique
While the temporal masks are novel and qualitatively "make sense" (though this is subjective), the generation of the spatial masks is not novel, is unconnected to the temporal mask generation, and often doesn't make sense because of lack of temporal smoothness / cohesion, particularly for C-LSTM, where the visualizations when the mask is on appear quite "jumpy" (see Fig 2, Seq 1). It would be great to see more innovation on the temporal mask generation to address some of these issues (one natural approach that comes to mind would be learn spatial masks as done in Fong & Vedaldi, 2017, possibly with a temporal smoothness term between spatial masks and possibly combining temporal + spatial masks for freezing operation, i.e., only freeze spatial pixels) -- that said, I realize that this may be out of scope for a rebuttal.

2. Lack of support for qualitative claims
The paper makes 3 claims based on qualitative examples shown (see above bullets in summary); these claims could be easily substantiated qualitatively (by evaluating over the dataset or a subset of it).

3. Lack of discussion on limitations/benefits of technique + how to use/interpret technique
* There are no baseline comparisons for the proposed temporal mask generation. natural ones would be visualizing saliency methods (i.e., gradient, SmoothGrad, occlusion [Zeiler & Fergus, 2014], RISE [BMVC 2018], for instance, as examples of a few easy-to-implement, representative methods for backprop and perturbation methods) w.r.t. to temporal dimensions and then thresholding and applying those baseline temporal masks and demonstrating that the proposed temporal mask generation is.
* There's no discussion (or experiments) on the benefits & limitations of their approach; this is important as we've seen from papers like Mahendran & Vedaldi, ECCV 2016, Adebayo et al., NeurIPS 2018 and Kindermans et al., arXiv 2017 that some saliency methods fail to meet basic desirata (i.e., specificity to output class and model weights, etc.). Ideally, the authors would show that their temporal mask generation meets some desired criteria (as well as compare with baseline methods) to justify their approach.
* One limitation of Fong & Vedaldi 2017 is the difficulty of finding global optimum for the different hyper-parameter terms (Fong et al, ICCV 2019 addresses this, which might be of interest to the authors). There's no discussion about whether this problem persists for this work. Also -- it'd be interesting to see whether the reverse loss function (i.e., maximizing class score) yields similar results.
* More discussion can be added about how to interpret results / use the technique (i.e., what is this technique useful/not useful for? what do results mean?). For instance, the claim that I3D temporal mask is shorter suggests a complex phenomenon -- that the necessary temporal evidence is smaller. This is a bit surprising to me, as I3D performs better overall, so I would have expected it to encode more redundancy (this can be checked by exploring the classes in which I3D does perform better) -- however, this interpretation differs from that of the authors ("This is especially visible in the temporal mask of Sequence #3, where it is active specifically").

In addition to responding to the above with relevant text + preliminary experimental results, I also had the following questions/asks:
* For misclassified classes in Fig 2, are you optimizing w.r.t. the top predicted class or the ground truth class? Do they differ substantially when optimizing for different output classes?
* What were the lambda hyperparameters from Eq. 1 (and how were they chosen)? Are these relatively stable or are their instances of technique failure due to the difficulty in balancing these terms?
* Show Table 1 on only the classes that were focused on (i.e., the ones w comparable performance between the two models); I'm wondering if the impact of reversal on I3D is less than that on C-LSTM, as claimed by the authors (it's hard to tell when their baseline performance is different)


**Experience Assessment:**

I have published one or two papers in this area.

**Review Assessment: Checking Correctness Of Derivations And Theory:**

N/A

**Review Assessment: Checking Correctness Of Experiments:**

I assessed the sensibility of the experiments.

**Review Assessment: Thoroughness In Paper Reading:**

I read the paper at least twice and used my best judgement in assessing the paper.

---

> ### Author Response · Authors · 2019-11-12
> **Reply to Official Blind Review #5, Part 1/2**
>
> Dear reviewer,
>
> We are thankful for your very thorough and constructive review. We have updated the article including support of the three qualitative claims that you have listed in bullet points (point #2 in your list). These numbers can be found in the new Table 2 in the article, confirming the difference between the two models. The discussion has also been improved with references to this analysis.
>
> We evaluated the three claims on a subset of the Something-something validation set; we ran the temporal mask optimization and Grad-CAM analysis for 791 samples (150 samples for each of the seven classes presented in that section, fewer when there were less than 150 samples present in the validation set). The masks take a few minutes per clip to optimize on one GPU meaning that for the scope of this rebuttal we were limited to this subset. For a final version of the article we would like to run this for all eleven classes under consideration and for more than 150 samples per class when available.
>
> As for your first point, we agree that this is a good idea that we will keep in mind for future work. Unfortunately, as you point out, it was out of the scope for the rebuttal time window.
>
> Coming to your third point, we first of all agree that especially the work by Adebayo et al. (NeurIPS 2018) is highly relevant for our article and are we are again thankful for this suggestion. We have now included it in the related work section, pointing out that Grad-CAM is one of the saliency methods that ‘passed’ in the experiments conducted by Adebayo et al.. We furthermore would like to stress that it cannot be the foregrounds of the images themselves that are the cause of the saliency maps (as it is for certain revealed-to-be edge detecting methods by Adebayo et al.), since we observe a significant difference between the saliency maps of the C-LSTM and those of the I3D (both computed with Grad-CAM) (Table 2). Concerning the sanity of the temporal mask method, we also observe a significant difference between the masks generated from the two different models. Although the latter is not an exhaustive check, this tells us that the temporal masks are not independent of the model, as was the case for several of the less sane methods listed in Adebayo et al. (whose results were unchanged when the weights of a network were random).
>
> Continuing on the third point, the suggestion of running baseline comparisons with temporal masks created from spatial saliency methods is a very constructive idea and something that would be interesting in order to motivate the separate optimization of our temporal mask. At the same time, extending the existing saliency methods to the temporal dimension by thresholding would be difficult, since the threshold would be difficult to choose. For instance, if we would base it on the magnitude of the saliency maps per frame, this would ultimately not work for clips with very small relevant spatial patches, and the threshold would be difficult to set. Thus, this is something we would want to think more about and instead suggest for future work.
>
> (This reply will continue in part 2, see below.)

---

> > ### Author Response · Authors · 2019-11-12
> > **Reply to Official Blind Review #5, Part 2/2**
> >
> > Next, we will discuss the hyperparameters of our proposed temporal mask method. These are $\lambda_1$ and $\lambda_2$ in Equation 1, the choice of optimizer and its learning rate, and the number of iterations for the optimizer. For the optimizer, we only used Adam and its standard (in Tensorflow and Pytorch) 0.001 learning rate. We tried both 100 and 300 iterations but decided to run for 300 iterations to let the masks have a fair chance of converging for both models. For time and computational reasons, we did not try a higher number of iterations than 300. Importantly, for the lambdas of Eq. 2, these do indeed affect the character of the masks. These were set as $\lambda_1, \lambda_2 = 0.01, 0.02$ for Something-something, and $\lambda_1, \lambda_2 = 0.02, 0.04$ for KTH. The reason that we penalized long and non-contiguous masks more for KTH was that a typical KTH sample contained four or five repetitions without pause of one action, and we wanted the mask to focus on well-defined parts of these actions.
> >
> > When starting to run our experiments, we found that when lowering $\lambda_1$, the masks became longer and when lowering $\lambda_2$ they became more fragmented. Having many hyperparameters to tune is not ideal, but hard to avoid. Since we found significant differences between the two models across the same set of hyperparameters per dataset we deemed the comparison to be fair and the masks to be informative. Previously, we referred to the public code repository for additional hyperparameters but we have now extended the hyperparameter section of the Appendix to include this important information and we would like to thank you for pointing this out so we could improve the transparency of our paper. Last, we thank the reviewer for another interesting suggestion: investigating whether the reverse loss function would yield stable results for one set of hyperparameters. Unfortunately we did not have time to run this during the rebuttal period and leave this for future work.
> >
> > In the reviewer’s fourth point, the observed phenomenon of the shorter temporal masks of I3D is brought up and problematized. Here, we take the chance of the OpenReview format to expand on how we interpreted this result. It is true that I3D overall has a stronger classification performance on the Something-something dataset. However, in our minds, this does not necessarily mean that this model does the most sensible feature extraction from the dataset. 3D CNNs are efficient learners with a vast number of parameters, and they can quickly find the most efficient patterns of the dataset with regard to the classification accuracy. If there is an efficient bias to learn, a 3D CNN will find it. It has previously been pointed out that supervised models in general in essence learn how to “cram for the exam” (e.g. Prof. Alexei A. Efros, “Self-supervision, Meta-supervision, Curiosity: Making Computers Study Harder” https://business.facebook.com/academics/videos/1632981350086599/), since there will likely always be some amount of dataset bias present. Although both of our compared models are within the supervised regime, it is informative to show what kinds of features or patterns (or biases) are learned by the models at comparable performances.
> >
> > As for the more general comment in the fourth point about adding more discussion and interpretation of the results, we find that our new results in Table 2 and the improved and updated discussion thereof have moved the article in this direction. Due to space constraints, we are not able to include more discussion in the main article. We however think that it is positive that the discussion on this page is public so that there is chance for a continued exchange regarding these questions, improving the conditions for future work.
> >
> >
> > Last, we address the shorter questions at the end of the review:
> >
> > * We are optimizing w.r.t. the predicted class (last sentence of Sect. 3.2)
> >
> > * The lambda hyperparameters were $\lambda_1, \lambda_2 = 0.01, 0.02$ for Something-something and $\lambda_1, \lambda_2 = 0.02, 0.04$ for KTH  (see the above paragraph about hyperparameters in this response, and see the added hyperparameter table for the temporal masks in the updated Appendix.)
> >
> > * The performance of the two models on the classes that were focused on in the article are listed in the first paragraph of Sect. 5.2 and in the beginning of Sect. A.2 for the remaining four classes. (Eleven classes in total were studied).
> >
> >
> > Please let us know if you wonder about any further matters that we could clarify.

---

### Decision · Program_Chairs · 2019-12-19

**Decision:**

Reject

**Comment:**

The paper addresses interpretability in the video data domain. The authors study and compare the saliency maps for 3D CNNs and convolutional LSTMs networks, analysing what they learn, and how do they differ from one another when capturing temporal information. To search for the most informative part in a video sequence, the authors propose to adapt the meaningful perturbations approach by Fong & Vedaldi (2017) to the video domain using temporal mask perturbations.
While all reviewers and AC acknowledge the importance and potential usefulness of studying and comparing different generative models in continual learning, they raised several important concerns that place this paper below the acceptance bar:
(1) in an empirical study paper, an in-depth analysis and insightful evaluations are required to better understand the benefits and shortcomings of the available and proposed models (R5 and R2). Specifically:
(i) providing a baseline comparison to assess the benefits of the proposed approach -- please see R5’s suggestions on the baseline methods;
(ii) analyzing how the proposed approach can elucidate meaningful differences between 3D CNNs and LSTMs (R5, R2). The authors discussed in their rebuttal some of these questions, but a more detailed analysis is required to fully understand the benefits of this study.
(2) R5 and R2 raised an important concern that the temporal mask generation developed in this work is grounded on the generation of the spatial masks, which is counterintuitive when analysing the temporal dynamics of the NNs - see R5’s suggestions on how to improve.
Also R5 has raised concerns regarding the qualitative analysis of the Grad-CAM visualizations. Happy to report that the authors have addressed these concerns in the rebuttal, namely reporting the results in Table 2 and providing an updated discussion. R1 has raised a concern about the importance of the sub-sampling in the CNN framework, which was partially addressed in the rebuttal.
To conclude, the AC suggest that in its current state the manuscript is not ready for a publication and needs a major revision before submitting for another round of reviews. We hope the reviews are useful for improving and revising the paper.